# The Prevalence of Autistic Traits in a Sample of Young Adults Referred to a Generalized Mental Health Outpatient Clinic

**DOI:** 10.3390/diagnostics14212418

**Published:** 2024-10-30

**Authors:** Irene Folatti, Giulia Santangelo, Claudio Sanguineti, Sanem Inci, Raffaella Faggioli, Angelo Bertani, Veronica Nisticò, Benedetta Demartini

**Affiliations:** 1Dipartimento di Scienze della Salute, Università degli Studi di Milano, 20142 Milan, Italy; irene.folatti@unimi.it (I.F.); giulia.santangelo@unimi.it (G.S.); claudio.sanguineti@unimi.it (C.S.); saneminci02@gmail.com (S.I.); 2Unità di Psichiatria 52, Presidio San Paolo, ASST Santi Paolo e Carlo, 20142 Milan, Italy; rafbeans@gmail.com (R.F.); benedettademartini@gmail.com (B.D.); 3Centro Giovani “Ettore Ponti”, Dipartimento Salute Mentale e Dipendenze, ASST Santi Paolo e Carlo, 20142 Milan, Italy; angelo.bertani@asst-santipaolocarlo.it; 4“Aldo Ravelli” Research Center for Neurotechnology and Experimental Brain Therapeutics, Università degli Studi di Milano, 20142 Milan, Italy; 5Dipartimento di Psicologia, Università degli Studi di Milano-Bicocca, 20126 Milan, Italy

**Keywords:** autism spectrum disorder, autistic traits, subthreshold autistic traits, mental health, screening, young adults

## Abstract

Background/Objectives: The diagnosis of Autism Spectrum Disorders (ASD) is undergoing significant revisions, impacting prevalence estimates in the general population. Moreover, the rise of a dimensional perspective on psychopathology has broadened our understanding of autism, recognizing that subthreshold autistic features extend throughout the general population. However, there remains a limited understanding of the prevalence of ASD traits in individuals with psychiatric disorders, particularly in young adults, who are at an age where several mental health conditions emerge. The aim of this study was to evaluate the prevalence of ASD traits in a sample of young adults (18–24 years old) attending a generalized mental health outpatient clinic. Methods: A total of 259 young adult patients completed the self-report screening questionnaires Autism Quotient (AQ) and Ritvo Autism and Asperger Diagnostic Scale-Revised (RAADS-R). Results: A total of 16.2% of our sample scored above the cut-off in both scales; this percentage decreased to 13.13% when restricting the RAADS-R cut-off to >119, as suggested for clinical samples. The association with sociodemographic features is discussed. Conclusions: We argue that screening for autistic traits should be integrated into the assessment of young adults presenting with nonspecific psychiatric symptoms or psychological distress. Although there is ongoing debate over the use of self-report screening tools, a positive result on both the AQ and RAADS-R should prompt clinicians to pursue a comprehensive diagnostic evaluation using structured or semi-structured interviews.

## 1. Introduction

The diagnostic category of Autism Spectrum Disorders (ASD) encompasses various conditions sharing the common features of “persistent deficits in social communication and interaction across multiple contexts” and “restricted repetitive patterns of behavior, interests or activities” [1]. The modern Dimensional Psychopathology Perspective considers ASD as existing on a continuum, rather than as a categorical disorder. ASD ranges from severe cognitive, social, and emotional delay, often recognized and diagnosed in childhood, to a pole commonly defined as “High-Functioning Autism”, which implies the absence of intellectual disabilities, along with a selective impairment in understanding and responding to social cues. Despite the popular belief that those with “High-Functioning Autism” can effortlessly manage daily activities (or manage them even better than neurotypical individuals, when labeled as “genius”), this oversimplification disregards the genuine challenges faced by individuals with ASD [2]. Moreover, this perspective helps explain why autistic traits might appear subthreshold but still contribute to significant impairments in functioning.

In personal and social realms, few individuals with ASD live independently or maintain social relationships or secure employment, often resulting in poor mental health and overall quality of life [3,4,5,6,7]; however, some adults with ASD successfully achieve post-secondary qualifications, long-term employment, and independence and engage in social and romantic relationships [8,9,10]. Importantly, autistic traits in individuals with ASD without intellectual disabilities may go unnoticed until adulthood [11,12], as they tend to acquire a repertoire of coping strategies (known as “camouflaging”) which serves to mitigate the impact of autistic traits in their daily lives. A growing body of the literature suggests that camouflaging is more prevalent in neurodivergent females than males, as they manage (not without difficulties) to adapt to the neurotypical environment; hence, their autistic traits are more difficult to recognize and acknowledge from an external (and clinical) perspective [13]. Overall, camouflaging is ultimately being considered as a key part of the female autism phenotype [14,15]. Consequently, this adaptive strategy frequently results in the postponement of seeking specialized mental health consultations, and these individuals are usually driven to seek clinical attention because of anxious and depressive symptoms often experienced as a comorbidity but not for the autistic traits themselves [13]. As a matter of fact, it is worth mentioning that adults diagnosed with ASD show rates of medical and psychiatric comorbidities up to twice as high as those of other psychiatric patients, including anxiety and mood disorders, obsessive compulsive disorder (OCD), and personality disorders [12,16,17,18]. Attention-Deficit/Hyperactivity Disorder (ADHD) has been found to be one of the most common comorbid diagnoses in adults with ASD, with a pooled prevalence of 25.7% [17]. Comorbidity can be explained by shared pathophysiology (e.g., ADHD, epilepsy), as an epiphenomenon of autistic traits and the challenges that are faced in everyday life (e.g., anxiety, depression), or with common symptom dimensions or diagnostic criteria (e.g., OCD, schizoid and schizotypal personality disorders) [19,20]. Anxiety is another common comorbidity in individuals with ASD, with specific phobias and generalized anxiety disorder (GAD) being the most frequent. The prevalence of anxiety disorders overall in individuals with ASD is approximately 20–30%. Moreover, recent research has drawn attention to the higher risk of suicide among individuals with ASD, especially when anxiety, depression, or other psychiatric conditions are comorbid, ultimately recommending close monitoring and mental health interventions to reduce suicidal risk itself [21,22].

The broadening of the diagnostic criteria of the ASD diagnosis [23] has been helping clinicians in identifying borderline situations, which can explain the recent increase in the prevalence of ASD diagnoses especially amongst adults [13,24,25,26,27]. According to the UK National Institute for Health and Care Excellence guidelines (NICE [28], 2012—last updated 2021), clinical assessment for ASD is required when patients have a diagnosis of a nonspecified mental disorder, previous or current contact with mental health services, or even if just one of the two core DSM-5 symptoms of ASD is satisfied. The Autism Diagnostic Observation Schedule—second version [29]—and the Autism Diagnostic Interview-Revised (ADI-R) [30] are the recognized gold standard tools for the diagnosis of ASD [31]; however, recent studies suggest that an integrated approach with combined tools, especially in adulthood, should be implemented [16,32]. As for available self-report tools, the Autism Spectrum Quotient (AQ) [33] and the Ritvo Autism and Asperger Diagnostic Scale-Revised (RAADS-R) [34] are the most common screening measures, as most of the studies assessed their validity in discriminating between adults with ASD and healthy controls [20,35,36]. Brugha et al. [37] went a step further by comparing the effectiveness of both the AQ and RAADS-R questionnaires in adults with a psychiatric diagnosis (364 men and 374 women) recruited in two mental health services in England. Their findings suggested that while both tools can aid psychiatrists and psychologists in referring individuals for comprehensive ASD diagnostic assessments, they should not be used in isolation. The study highlighted that the AQ items were less internally consistent compared to those of the RAADS-R, but the AQ was easier to complete; notably, a higher percentage of participants in their sample failed to complete the RAADS-R than those that failed to complete the AQ (12% vs. 10%). Crucially, they also identified that the optimal threshold for the RAADS-R in clinical populations was significantly higher than the one suggested by its developers (120 vs. 65). Moreover, novel approaches in a cross-cultural and personalized perspective have been explored. There is growing interest in validating diagnostic tools across different populations, as cultural factors may influence how ASD is expressed and diagnosed. These approaches could enable more tailored treatments for ASD in the future [38].

Overall, considering the difficulties in reaching clinical attention due to camouflaging defense mechanisms, the high rates of psychiatric comorbidity they might encounter, and the challenges in formulating a diagnosis itself when presenting to a clinician for an apparently unrelated symptomatology, individuals with ASD traits, and especially young adults, are deemed to be at a higher risk of poorer mental health outcomes and functional impairment, compared to neurotypical subjects of the same age [39,40,41]. However, to the best of our knowledge, no study has specifically examined the prevalence of ASD traits in a clinical psychiatric sample of young adults, particularly those presenting with nonspecific symptoms such as anxiety or depression. While autism research has traditionally centered on childhood diagnosis, far less is understood about young adults who may have undiagnosed ASD traits becoming evident only when they face the heightened social and emotional demands of adulthood. Identifying autistic traits in this population could reveal cognitive processing difficulties that tend to emerge in social settings such as work or higher education settings. These individuals may struggle to comprehend and interpret others’ thoughts, emotions, and intentions, which is considered a core challenge in ASD and significantly impacts social communication and interaction. Understanding these dynamics helps explain the relationship between autistic traits and mental health challenges in young adults, supporting the need for early screening in clinical settings to better address the mental health needs of this population. Addressing this gap is crucial for both clinical practice and research, as it can prevent delayed diagnoses or misdiagnoses and improve mental health outcomes and overall quality of life. Current screening practices often overlook ASD traits, particularly in patients without typical symptoms. Utilizing validated tools such as the AQ and the RAADS-R can enable earlier detection and lead to more comprehensive mental health assessments. Furthermore, given the comorbidities discussed earlier, identifying ASD traits as early as possible is especially important in young adults, as behavioral interventions applied early can reduce problematic behavior in 80–90% of cases. Overall, closing this gap will enhance our understanding of how autistic traits manifest in adulthood and improve mental health care strategies for this group.

### Aims of This Study

This study’s main objective is to explore the prevalence of autistic traits in a clinical outpatient population of young adults aged 18 to 24 years old, using the two most common self-report screening questionnaires, namely the AQ and the RAADS-R. These questionnaires aim to fill the knowledge gap concerning ASD in young adults with psychiatric conditions and contribute to more informed diagnostic and treatment practices. To better characterize our sample, we also investigated and discussed the role of sociodemographic factors such as sex assigned at birth and gender identity.

This study’s findings might lead to broader implications for both people with autism and mental health professionals. For individuals with ASD, the early identification of autistic traits in young adults, particularly those with psychiatric conditions, might lead to improved mental health outcomes; as previously mentioned, young adults may experience significant distress from undiagnosed ASD traits, often leading to social, educational, and employment challenges. Mental health professionals could adopt in their initial screening validated tools such as the AQ and RAADS-R to flag potential autistic traits that may otherwise go unnoticed. This could lead to more comprehensive treatment plans that address both ASD traits and co-occurring mental health issues, such as anxiety or depression. Additionally, understanding the relationship between sociodemographic factors and ASD traits allows for more sensitive, individualized care. Based on the existing literature, policy initiatives could emphasize the importance of early screening and intervention for ASD traits in adulthood. This includes promoting access to ASD diagnostic services in general mental health settings, creating employment support programs for individuals with ASD, and improving the accessibility of mental health services for marginalized groups. Policymakers can also advocate for expanded training programs for mental health professionals to better recognize and address autistic traits, thus reducing the rate of misdiagnosis and improving long-term outcomes for individuals with ASD [42]. These implications highlight the need for an integrated approach in both clinical practice and public health policy, focusing on early intervention and individualized support for those with ASD traits in adulthood.

## 2. Materials and Methods

### 2.1. Participants

The study population comprised 259 young adult patients referring to the “Centro Giovani Ettore Ponti” (ASST Santi Paolo e Carlo, Milan, Italy), a specialized consultative clinic for individuals aged between 14 and 25 years old seeking help for unspecified psychiatric symptoms and psychological suffering. Exclusion criteria were as follows: (i) age less than 18 years old or (ii) inability to understand the instructions of the task. This study was conducted in accordance with the Declaration of Helsinki and approved by the Ethics Committee “Comitato Etico Territoriale Lombardia 1” (protocol code CET 279-2024, 17 July 2024) for studies involving humans. Participants were told in clear language that this study consisted of compiling two self-report questionnaires that would be scored for both clinical and research purposes; it was made clear that should the questionnaires indicate the presence of autistic traits, the participants would be evaluated further with clinician-administered tools and a further clinical interview; at the same time, it was explained that data would be anonymized and shared amongst professionals bound by confidentiality within our hospital and always maintaining anonymity. It was stated that this study was approved by the local ethics committee and that participants, even after giving informed consent, were free to withdraw from this study at any time without giving further explanation. Thereafter, informed consent was obtained.

### 2.2. Procedure

First, participants engaged in an in-person consultation with a clinician, as a usual part of the diagnostic assessment at the “Centro Giovani Ponti”. Sociodemographic information was collected, such as sex assigned at birth, height, weight, gender identity, education level, employment, and living environment. During this consultation, participants were provided with the questionnaire used in this study, and responses were collected through an online questionnaire due to the concomitant COVID-19 pandemic. They received a QR code, which, when scanned, automatically redirected them to an online form containing the Italian versions of the following questionnaires: (i) The AQ was used, where a Total Score ≥ 32 suggests the presence of ASD traits. Five subscales were also used according to the authors’ instructions: Social skills, Attention switching, Attention to detail, Communication, and Imagination [33,43]. (ii) The RAADS-R was also used, where a Total Score > 65 suggests that the participant should be assessed further for ASD [34,44]; however, as mentioned in the Introduction, Brugha et al. [37] found that a cut-off of >119 has a better accuracy in a sample of psychiatric patients; hence, we analyzed our data considering both the suggested cut-offs. Moreover, four subscales were also used according to the authors’ instructions: Social Relatedness, Circumscribed Interests, Language, and Sensory–motor [34].

Participants were instructed to complete these questionnaires independently, at home, within 48 h of the consultation. They were provided with the clinician’s contact information for any questions potentially emerging during the questionnaire completion process. At the end of the first consultation, before receiving the results of the questionnaire, the clinician categorized each patient’s symptomatology according to the following categories: anxiety symptoms; personality disorder symptoms; mood disorder symptoms; potential disorders with onset during childhood or neurodevelopmental conditions; psychotic symptoms; potentially disturbed eating behaviors with clinical significance; or multiple symptoms to be further investigated. Subsequently, a thorough diagnostic assessment of general psychiatric conditions was conducted following DSM-5-TR criteria by a psychiatrist and a psychotherapist, by means of clinical interviews conducted after the first contact with the mental health service. Moreover, all cases, including the ones presented in this study, are usually discussed in Equipe Meetings, together with the presence of nurses, clinical psychologists, and social assistants.

### 2.3. Statistical Analysis

Statistical analyses were performed using Statistical Package for Social Sciences (SPSS) version 27; a significant threshold was set at α ≤ 0.05, and all tests were two-tailed. There were no missing values in the dataset.

First, the Kolmogorov–Smirnov test was applied to verify that each variable followed a normal distribution. Descriptive statistics were calculated for sociodemographic and clinical variables. We analyzed the frequencies of patients scoring above the cut-off on the screening questionnaires, with participants considered at risk for ASD if they scored above the cut-off on both the AQ and the RAADS-R. Contingency tables and χ^2^ tests were used to evaluate the distribution of participants scoring above the cut-off on both the AQ and the RAADS-R according to their sex assigned at birth and gender identity. Lastly, *t*-tests for independent samples were conducted to examine differences in the AQ and RAADS-R scores for male and female participants (based on sex assigned at birth).

## 3. Results

Of the 259 patients assessed and included in our sample, none already had a diagnosis of ASD before being admitted to our clinic (see Table 1 for sociodemographic features).

We had no missing data with respect to the AQ and the RAADS-R. Five participants decided to not declare their gender for personal reasons and were considered in the group “Undeclared” (as it does not represent missing data but a specific choice of the participant).

The mean AQ Total Score was 22.08 (±7.94); the mean RAADS-R Total Score was 82.33 (±45.21). A total of 42 (16.22%) participants scored above the cut-off on the AQ, and 149 (57.53%) subjects scored above the cut-off on the RAADS-R (analyzed with the standard cut-off > 65). All 42 participants who were positive on the AQ were positive also on the RAADS-R, while 107 patients were positive on the RAADS-R but not on the AQ; no participant was positive on the AQ but not on the RAADS-R. Finally, 110 (42.47%) subjects scored below the cut-off on both the AQ and RAADS-R. However, considering the cut-off of >119 for the RAADS-R, we found 55 (21.24%) participants scoring above the cut-off, and there were 34 (13.13%) participants scoring positive on both the AQ and RAADS-R.

Sex assigned at birth had a significant effect on the Total Score of the RAADS-R > 65 (χ^2^(1) = 6.423, *p* = 0.026). Individuals with female sex assigned at birth scored significantly lower than individuals with male sex assigned at birth on the AQ Imagination subscale (t = −2.829, *p* = 0.005) and on the RAADS subscales Social Relatedness (t = −2.209, *p* = 0.028) and Language (t = −2.646, *p* = 0.009) (Table 2).

Gender identification was associated with the Total Scores of all screening questionnaires: AQ (χ^2^(3) = 8.392, *p* = 0.039); RAADS > 65 (χ^2^(3) = 13.918, *p* = 0.003); and RAADS > 119 (χ^2^(3) = 16.975, *p* = 0.001) (Table 3).

Overall, the results show that screening tools like the AQ and the RAADS-R in this context provide valuable insights for clinicians, highlighting the importance of incorporating autism spectrum assessments in routine evaluations of young adults showing psychiatric symptoms.

## 4. Discussion

In this study, we aimed to assess the prevalence of autistic traits in a group of 259 young adult patients referred to a psychiatric outpatient clinic. In our sample, the AQ and the RAADS-R showed markedly different results: 42 (16.22%) subjects scored positive om the AQ and 149 (57.53%) on the RAADS-R using the standard cut-off of >65, so 42 (16.2%) subjects scored positive on both scales. Interestingly, when setting the RAADS-R cut-off at >119 (as in Brugha et al., 2020) [37], 55 (21.24%) participants scored above it, thus being closer to the percentage of those scoring above the cut-off of the AQ; in this case, 34 (13.13%) participants scored positive on both measures. This prevalence seems considerably high, although it needs to be further contextualized. First, the prevalence rates of adult individuals fully diagnosed with ASD in the general population vary from 9.8 to 11 per 1000 in the UK [45,46] and from 2.21 to 3.66 per 1000 in the US [47,48]. Second, existing evidence of ASD prevalence and autistic traits in the clinical population, and especially in individuals who sought mental health support for symptoms apparently not strictly related to ASD (e.g., anxious and depressive symptoms), is still scarce: a wide range of prevalence rates, from 1.3% up to 18.9%, are reported in clinical samples (see Table 4 for further information). These discrepancies between different studies might be imputable to the different data sources used (such as diagnostic assessment, medical records, and/or national health statistical data) and to differences between outpatient clinics themselves [49].

Overall, our prevalence rate of 13.3% of patients scoring above the cut-off on both the AQ and RAADS-R (considering the most restrictive proposed cut-off) seems to fall at the higher end of the aforementioned range of ASD prevalence in clinical populations. It must be noted that our sample is somehow unique: the “Centro Giovani Ponti” receives young individuals who, spontaneously or upon suggestion by a significant other (e.g., relative, a friend, or another specialist), seek help for unspecified psychiatric and psychological suffering. The present assessment for ASD was conducted in the very first phases of patients’ care, when a proper diagnosis had not been reached yet. To the best of our knowledge, this is the first study assessing this specific kind of population and in such a strict range of age (18–24 years old), an age that deserves increased attention since, given the progressive separation from protective environments such as school or home family, subthreshold and/or hidden symptoms of ASD might be exacerbated [54]. Few studies have been conducted on ASD and autistic trait prevalence in the general young population. Adachi and colleagues (2020) [55] found that, amongst a group of 37 university students who spontaneously requested psychological counseling, 9 participants scored above the cut-off on the AQ, accounting for a percentage of 24.3% (slightly higher than the one we found); in their control group, composed of 68 students who never requested psychological counseling, only 3 subjects (4.4%) scored above the cut-off. In a study conducted by Eberhard et al., 2022 [56], the prevalence of neurodevelopmental disorders (NDDs, specifically ADHD and ASD) was investigated in 170 young adults aged 18 to 25 years, who attended for the first time an adult outpatient psychiatric (AOP) non-psychosis clinic in Stockholm. A total of 49 patients (25%) had already formally received a diagnosis of psychiatric disorder/NDD in previous contact with child or adult psychiatric clinics. Two ASD screening tools were used: the Autism Symptom SElf-ReporT for adolescents and adults (ASSERT) [57] and the Adult Autism Spectrum Quotient (AQ) [58]. ASD was diagnosed in 34 patients (20%), 18 of whom (53%) also met the criteria for ADHD. A total of 7 of the 34 (21%) had been diagnosed with ASD in childhood. Overall, more than two-thirds of the subjects were diagnosed with a type of NDD, and the majority had not been diagnosed in childhood. The authors also underlined that the NDD group usually met criteria for other psychiatric disorders (affective anxiety, OCD spectrum, or personality disorder). The high rate of comorbidity could mask the presence of NND symptoms, delaying diagnosis in young adults. The authors point to the limitations of their study such as the exclusion of patients with psychotic disorders, the limited age range, and the impossibility of generalizing the conclusions to other outpatient populations. Other works on young adults recruited from the general population showed a much lower trait prevalence than the one we found in our sample: in a group of volunteer college students in the U.S., 1.9% scored above the cut-off for the AQ [59], while in a large sample of 752 high school students in southern Italy, only 0.8% did [60].

### 4.1. Sociodemographic Features

When addressing sex and gender differences, first, we found a significantly different distribution between males and females (according to the sex assigned at birth) only using the cut-off of 65 but not with the cut-off of 119 on the RAADS-R. Males scored higher than females on the AQ subscale Imagination, in line with a recent meta-analysis investigating sex differences in AQ subscales in non-clinical samples [61]; moreover, the meta-analysis showed that males scored higher in the Imagination subscale (i.e., had a less developed imagination ability, considered as “forming new ideas, mental images, or concepts”) than in any other AQ subscale. Considering these results, the authors suggested that imagination represents the facet of autism that best accounts for the strongly male-biased sex ratio (i.e., the higher frequency of ASD diagnoses in males), given that reduced imagination represents a criterion for ASD diagnosis [61]. As for RAADS-R subscales, we found that males scored higher than females in the subscales Language and Social Interaction, again consistent with the published literature (for a meta-analysis, see Moseley et al., 2018 [62]). We did not find a significant difference in the RAADS-R Restricted Interest subscale, despite the fact that it was previously described that women with ASD, in particular those without intellectual disabilities, display lower restricted and repetitive behaviors or interests than men [63]. Second, with respect to gender identity, we found that the group of individuals declaring themselves “non-binary” scored significantly higher than the ones declaring themselves males or females on both the AQ and the RAADS-R (Table 3). The literature on this specific topic is very poor: in one study, Strang and colleagues [64] found evidence of an increased gender variance since childhood in children diagnosed with ASD compared to children without ASD; in a more recent study, Kung [65], by administering the AQ to 323 non-clinical adults who declared themselves to belong to gender minorities, found that transgender men and non-binary individuals assigned female at birth scored significantly higher on the AQ than both female participants and non-binary people assigned male at birth. Although preliminary and to be interpreted with caution due to the non-homogeneous distribution of the sample (only 12 non-binary individuals), our findings seem to be in line with the literature, and we believe that this theme deserves further consideration.

As for living conditions, most of our sample (232, 89.58%) was living with their parents, 15 (5.79%) were living with roommates, 5 (1.93%) were living alone, 5 (1.93%) with their partner, and 2 (0.77%) in a therapeutic community. Our results are in line with the published literature: among adults with higher-functioning ASD, the rates of independent or semi-independent living range from 16 to 36% [4,10], with only one study reporting a rate of 50% in a sample of 18 adults with diagnoses of High-Functioning Autism and Asperger’s Disorder [66]. Finally, considerations about the sensitivity and specificity of the questionnaires we implemented should be made. The choice of a new cut-off of >119 for the RAADS-R was suggested by Brugha et al. (2020) [37] in a cohort-design study that evaluated AQ and RAADS-R diagnostic accuracy in a clinical population composed of 738 adult individuals from outpatient and inpatient mental health clinics where the most common diagnoses were mood disorders (primarily bipolar affective disorder (*n* = 39; 23% of mood disorders) and depression (*n* = 62; 37% of mood disorders)); with respect to the RAADS-R, they found a sensitivity equal to 0.75 (95% CI [0.48, 0.93]) and a specificity equal to 0.71 (95% CI [0.60, 0.81]), compared to the 97% sensitivity and 100% specificity of the validation study that set the cut-off at 65, although in the general population [34]. They pointed out that both the AQ and RAADS-R were validated through a case–control design study, which has been declared not recommended for test validation because of the risk of overestimation [67]: in fact, according to the authors, a cohort design would be more appropriate for validation studies, since the sample of “yet-to-diagnose” subjects is more representative of the population in which the test is going to be used. In this study, since we did not have a confirmed diagnosis yet, we were not able to assess the accuracy of the RAADS-R with this new cut-off; however, given the similar ratio of positive individuals on the AQ and the RAADS-R when using the >119 cut-off, we believe that our results are in line with the work by Brugha et al., suggesting that “119” could be the recommended threshold for the RAADS-R in general psychiatric clinical samples and more specifically in young adult patients.

### 4.2. Clinical, Theoretical, and Policy Implications

The implications of our findings center on several key areas related to early identification and intervention for young adults with autistic traits.

First, early detection can lead to timely interventions, potentially reducing the negative impact of untreated ASD symptoms. We advocate for the routine use of screening tools such as the AQ and the RAADS-R in mental health settings to identify individuals who may have been overlooked in previous evaluations. Early identification is essential for designing personalized interventions and providing better access to resources, such as vocational training and social skill programs. These resources can help young adults to navigate critical life stages, ultimately preventing symptom escalation and improving the overall quality of life and well-being.

Second, from a theoretical perspective, identifying psychiatric symptoms co-occurring with autistic traits can refine our understanding of how neurodevelopmental and psychiatric conditions interact. For instance, these overlapping symptoms may suggest that, in a transitional life stage like young adulthood, difficulties in executive function might exacerbate psychiatric conditions.

Finally, from a policy standpoint, our results emphasize the need for mental health systems to adopt more comprehensive screening protocols for autism in young adults presenting psychiatric symptoms. Spectrum-based approaches can guide the development of more inclusive diagnostic tools and interventions that address both autism and mental health needs. Creating supportive environments that recognize a spectrum of neurodiverse identities, both in everyday social contexts and mental health services, is crucial to ease the transition from pediatric to adult outpatient care.

### 4.3. Limitations and Future Directions

We acknowledge the limitations of our study. First, our participants independently completed the self-report questionnaires, and the diagnostic confirmation of ASD through a clinical or semi-structured interview is lacking; hence, inferences about actual ASD prevalence cannot be made. Second, since our sample did not receive a specific psychiatric diagnosis, we could not stratify the participants according to their major psychiatric symptomatology, nor could we assess other potential psychiatric comorbidities or their possible association with ASD traits.

Future research could aim to refine diagnostic tools like the AQ (which was validated on a sample of male individuals only) and the RAADS-R for specific populations (e.g., women, people with psychiatric comorbidities). Developing more sensitive and specific screening measures for populations that often go undiagnosed could help bridge current gaps in clinical practice. An essential point should be testing the reliability of these results using other tools administrated by trained and expert clinicians. Second, a follow-up with clinician-administered tools such as the ADOS-2 would be crucial to test the incidence of ASD-diagnosed people in this wide group of individuals with autistic traits. Finally, future research would benefit from longitudinal designs: by following individuals over time, researchers can gain insights into how autistic traits and psychiatric symptoms co-evolve and how effective early interventions are.

## 5. Conclusions

In conclusion, we showed that, in a population of young adults (who are at a critical age for the outbreak of psychiatric diseases) referred to a dedicated psychiatric outpatient clinic, the rate of ASD traits is not negligible. Therefore, despite the non-unanimous consensus over the self-report tools used in this study, we argued that positivity on both the AQ and RAADS-R, with the recently proposed cut-off of 119, indicates that a subsequent full assessment with structured or semi-structured interviews is needed.

## Figures and Tables

**Table 1 diagnostics-14-02418-t001:** Sociodemographic features.

Variable		Value
Age, mean (SD)		19.63 (1.98)
Gender identity, *N* (%)	Female	156 (60.23)
	Male	86 (33.20)
	Not binary	12 (4.63)
	Undeclared	5 (1.93)
Education, *N* (%)	Middle school (8th grade)	91 (35.14)
	3-year professional license	19 (7.34)
	Diploma	139 (53.67)
	Bachelor’s degree	9 (3.47)
	Master’s degree	1 (0.39)
Employment, *N* (%)	Student	195 (75.99)
	Employed	34 (13.13)
	Unemployed	30 (11.58)
Living condition, *N* (%)	Living alone	5 (1.93)
	Living with parents	232 (89.58)
	Living with partner	5 (1.93)
	Living with roommates	15 (5.79)
	Living in a therapeutic community	2 (0.77)
Clinician’s preliminary report after the first consultation, *N* (%)	Anxiety symptomatology	104 (40.15)
	Personality disorder symptomatology	56(21.62)
	Mood disorders	26 (10.04)
	Disorders with onset during childhood or neurodevelopmental condition	8 (3.09)
	Psychotic symptoms	4 (1.54)
	Eating behavior-related symptoms	3 (1.16)
	Multiple symptoms to be further investigated	54 (20.85)

Abbreviations: *N* = numerosity; SD = Standard deviation; % percentage.

**Table 2 diagnostics-14-02418-t002:** Psychometric assessment.

		Overall Value	Comparison Between Females and Males *				
			F (*N* = 171)	M (*N* = 88)	*t* or χ^2^	*p*	Cohen’s *D*
AQ Total Score, mean (SD)		22.08 (7.94)	21.67 (7.95)	22.88 (7.92)	−1.160	0.247	−0.152
AQ Total Score, *N* (%)	Below cut-off	217 (83.78)	143	74	0.009	0.923	NA
	Above cut-off	42 (16.22)	28	14			
AQ Social skills, mean (SD)		4.20 (2.62)	4.16 (2.73)	4.26 (2.41)	−0.284	0.777	−0.037
AQ Attention switching, mean (SD)		5.98 (2.19)	5.91 (2.21)	6.11 (2.14)	−0.722	0.471	−0.095
AQ Attention to detail, mean (SD)		4.77 (2.38)	4.75 (2.37)	4.81 (2.42)	−0.167	0.867	−0.022
AQ Communication, mean (SD)		3.73 (2.45)	3.68 (2.43)	3.83 (2.5)	−0.452	0.652	−0.059
AQ Imagination, mean (SD)		3.40 (1.93)	3.16 (1.89)	3.86 (1.931)	−2.829	0.005	−0.371
RAADS-R Total Score, mean (SD)		82.33 (45.21)	79.63 (46.04)	87.58 (43.315)	−1.343	0.180	−0.176
RAADS-R cut-off >65 Total Score, *N* (%)	Below cut-off	110 (42.47)	81	29	4.94	0.026	NA
	Above cut-off	159 (57.53)	90	59			
RAADS-R cut-off >119 Total Score, *N* (%)	Below cut-off	204 (78.76)	133	71	0.293	0.588	NA
	Above cut-off	55 (21.24)	38	17			
RAADS-R Social Relatedness, mean (SD)		39.98 (21.40)	37.89 (21.58)	44.05 (20.57)	−2.209	0.028	−0.290
RAADS-R Circumscribed Interests, mean (SD)		16.82 (10.16)	16.27 (9.84)	17.88 (10.74)	−1.201	0.231	−0.158
RAADS-R Language, mean (SD)		6.10 (4.64)	5.56 (4.57)	7.15 (4.63)	−2.646	0.009	−0.347
RAADS-R Sensory–motor, mean (SD)		19.43 (14.52)	19.91 (14.979)	18.51 (13.61)	0.732	0.465	0.096

* According to sex assigned at birth. Abbreviations: AQ = Autism Quotient; *N* = numerosity; RAADS-R = Ritvo Autism Asperger Diagnostic Scale-Revised; SD = Standard deviation. Degrees of freedom for all comparisons between continuous variables: 257. Degrees of freedom for all comparisons between categorical variables: 1.

**Table 3 diagnostics-14-02418-t003:** Distribution according to self-reported gender identity.

		AQ		RAADS-R > 65		RAADS-R > 119	
	*N*	Below Cut-Off	Above Cut-Off	Below Cut-Off	Above Cut-Off	Below Cut-Off	Above Cut-Off
Female	156	134 (85.9%)	22 (14.1%)	79 (50.64%)	77 (49.36%)	126 (80.77%)	30 (19.23%)
Male	86	73 (84.88%)	13 (15.12%)	29 (33.72%)	57 (66.28%)	71 (82.56%)	15 (17.44%)
Non-binary	12	7 (58,33%)	5 (41.67%)	2 (16.67%)	10 (83.33%)	4 (33.33%)	8 (66.67%)
Not declared	5	3 (60%)	2 (40%)	0 (0%)	5 (100%)	3 (60%)	2 (40%)
Statistical analysis		χ^2^(3) = 8.392, *p* = 0.039		χ^2^(3) = 13.918, *p* = 0.003		χ^2^(3) = 16.975, *p* = 0.001	

Abbreviations: AQ = Autism Quotient; *N* = numerosity; *p* = significance threshold (α = 0.05); RAADS-R = Ritvo Autism Asperger Diagnostic Scale-Revised; % percentage.

**Table 4 diagnostics-14-02418-t004:** ASD prevalence (i.e., full diagnosis) rates in clinical samples.

Author, Year	Prevalence Rate	Clinic, Country	Assessment Methods
Nylander, 2001 [50]	1.4%	General outpatient mental health service, Sweden	Administration of “Autism Spectrum Disorder in Adults Screening Questionnaire”, analysis of psychiatric records, and parent interview with Asperger Syndrome Screening Questionnaire (ASSQ) and ASDI.
Nylander et al., 2013 [51]	1.3%	Psychiatric services of a university hospital, Sweden	Retrospectively checked ASD (or equivalent) diagnosis among accesses in hospital from past 20 years. No information is known about screening and diagnostic evaluation.
Takara et al., 2014 [52]	16%	Adult outpatient clinic for depression, Japan	AQ with cut-off of 26 (declared to be optimal one for Japanese version in clinical samples—Sn 0.76 and Sp 0.71), and diagnostic confirmation through interview with expert.
Brugha et al., 2020 [37]	4.8%	Adult mental health service (inpatients and outpatients), UK	Administration of AQ and RAADS-R for screening purposes (exact numbers of patients who scored positive on one or both scales is not reported); then, subjects were diagnosed using ADOS module 4.
Nyrenius et al., 2022 [53]	18.9%	Adult psychiatric outpatient clinic, Sweden	Administration of “RAADS-14 Screen” and “20 items Asperger Syndrome Diagnostic Interview”

Abbreviations: ADOS = Autism Diagnostic Observation Schedule; ASD = Autism Spectrum Disorder; AQ = Autism Quotient; RAADS = Ritvo Autism Asperger Diagnostic Scale; % percentage.

## Data Availability

Due to ethical restrictions, anonymized data will be made available upon request to the corresponding author by any qualified researcher.

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
