# Peer review of "The Prevalence of Autistic Traits in a Sample of Young Adults Referred to a Generalized Mental Health Outpatient Clinic"

_diagnostics, 2024, doi:10.3390/diagnostics14212418_

Round 1
Reviewer 1 Report
Comments and Suggestions for Authors
The article is original in its conception and is of interest to a wide range of readers. There are a few small remarks
It is not clear in what language the questionnaires (the Autism Spectrum Quotient (AQ) [30] and the Ritvo Autism and Asperger Di- 85 agnostic Scale-Revised (RAADS‐R) [31]) were presented as the references were given only to English-language sources.
In the table 1 it is not clear what “Middle School 91 (35.14)” means.
I have concerns with data interpretation when there is such a huge difference in the number of subjects (employed 229 and unemployed 30). I would recommend to skip these data. The data on the sex difference should be interpreted with caution for the same reason.
“(χ(3)” is not widely used presentation - is it a chi square index?
Author Response
We thank the reviewer for their positive and helpful comments, which we hope have helped us improve our manuscript. For clarity, we addressed each comment one by one.
Reviewer 1
The article is original in its conception and is of interest to a wide range of readers. There are a few small remarks
Comment: It is not clear in what language the questionnaires (the Autism Spectrum Quotient (AQ) [30] and the Ritvo Autism and Asperger Diagnostic Scale-Revised (RAADS‐R) [31]) were presented as the references were given only to English-language sources.
Response: The Autism Spectrum Quotient (AQ) and the Ritvo Autism and Asperger Diagnostic Scale-Revised (RAADS‐R) are commonly available in multiple languages, including Italian. For formal use in Italy, translations of these questionnaires exist and are typically cited in local studies or validation papers.
For the RAADS-R, an official Italian version exists, translated by Davide Moscone and David Vagni, accessible through the official website https://www.spazioasperger.it/il-raads-r-la-scala-per-diagnosticare-spettro-autistico-lieve-e-sindrome-di-asperger-negli-adulti/. The authors translated and reviewed the Italian version of the RAADS-R scale, with back-translation supervised by Federica Vasta, a translator and interpreter, along with a native speaker. The Italian version available on the website is officially accepted by the original scale's authors (Ritvo et al.). They are currently working on a validation study for the Italian population. In the meantime, this translated version has been used in various clinical settings in Italy, ensuring its applicability to Italian-speaking population, and it is still considered an effective tool for diagnosing adults with suspected autism spectrum disorders (ASD), particularly for those with high-functioning autism or Asperger's syndrome. Its sensitivity and specificity have been confirmed in various international studies.
The Autism-Spectrum Quotient (AQ) has been validated for use with Italian-speaking population. A study conducted in 2012 by Ruta and colleagues confirmed that the Italian version of the AQ effectively captures different autism phenotypes (broader, medium, and narrow) in a Sicilian sample, demonstrating that the AQ is a cross-culturally reliable tool for assessing the severity of autistic traits in Italian-speaking populations and is useful in both clinical and research settings. (Ruta, L., Mazzone, D., Mazzone, L., Wheelwright, S., & Baron-Cohen, S. (2012). The broader autism phenotype in parents of children with autism: A Sicilian study using the Autism Spectrum Quotient. Journal of Autism and Developmental Disorders, 42(5), 625-633)
Relevant references were added to the manuscript.
Comment: In the table 1 it is not clear what “Middle School 91 (35.14)” means.
Response: The “Middle School” license in Italy correspond to the 8th grade license and refers to the level of education. By mistake, it was merged with the row “gender”. We corrected the typo and specified that Middle School equals to the 8th grade.
Comment: I have concerns with data interpretation when there is such a huge difference in the number of subjects (employed 229 and unemployed 30). I would recommend to skip these data. The data on the sex difference should be interpreted with caution for the same reason.
Response: We agree with the reviewer that the comparison between employed and unemployed participant is too unbalanced to be reliable; hence, we removed all that data from our manuscript (methods, results, and discussion paragraphs). However, we believe that the data related to sex and gender differences are relevant to the scope of our paper; hence, we kept the analysis in the manuscript but added a consideration as follows: “Although preliminary and to be interpreted with caution due to the non-homogeneous distribution of the sample (only 12 non-binary individuals), our findings seem in line with the literature, and we believe that this theme deserves further consideration.”
Comment: “(χ(3)” is not widely used presentation - is it a chi square index?
Response: it is a χ2 index, with, amongst parentheses, the degrees of freedom of the analysis. In this case, we compared four groups of gender identity, hence the degrees of freedom are 4 – 1 = 3.
To clarify this point, we amended the text to report χ2 instead of χ where applicable.
Reviewer 2 Report
Comments and Suggestions for Authors
Investigating the prevalence of autistic traits in young adults seeking mental health services is essential for improving diagnosis, treatment, and overall well-being for individuals with ASD. By addressing the unique needs of this population, we can promote a more inclusive and supportive mental health care system.
My suggestions for improving the manuscript are listed in the following:
Introduction
It effectively introduces the topic of autistic traits in young adults seeking mental health services, highlighting the key challenges and the importance of studying this population. The introduction is well-structured, concise, and engaging, making it a compelling starting point for the paper.
However, I think the introduction could be revised based on the following considerations:
-Review of the Literature:
The literature cited could be up-to-date, especially considering the rapid advancements in research on ASD and mental health.
Summarize the most relevant findings from previous studies to provide a solid foundation for the current research.
-Theoretical Framework:
Discuss the theoretical frameworks (e.g., cognitive, social-emotional) that inform the research question.
Clearly articulate how these theories relate to the study's objectives and contribute to our understanding of autistic traits in young adults.
-Research Gap:
Identify the specific knowledge gap that the study aims to address.
Explain why filling this gap is crucial for advancing research and improving clinical practice.
-Research Questions or Hypotheses:
State the research questions or hypotheses that the study will investigate.
Ensure that the research questions or hypotheses are directly related to the topics discussed in the introduction.
-Significance:
Emphasize the broader implications of the study's findings for individuals with ASD, mental health professionals, and policymakers.
Finally, I suggest highlighting potential benefits by discussing how the research may contribute to improved diagnosis, treatment, and outcomes for young adults with ASD.
Methodology:
The Materials and Methods section clearly and comprehensively describes the research procedures. It includes essential information on participants, procedures, and statistical analyses.
However, there are a few areas where the section could be improved:
-Ethical Considerations:
While the authors mention obtaining informed consent, it would be beneficial to provide more details about the consent process, such as the information provided to participants and the measures taken to ensure their understanding.
The section should explicitly address how participant confidentiality was maintained throughout the study.
-Clinical Assessment:
This section could provide more information about the specific diagnostic criteria clinicians use to categorize participants' symptomatology.
If multiple clinicians were involved in the assessments, the authors should discuss the measures taken to ensure inter-rater reliability.
-Statistical Power:
While the sample size is relatively large, a power analysis could be conducted to determine the study's ability to detect significant effects.
-Missing Data:
The authors should clarify how missing data were handled, such as imputation methods or exclusion criteria.
Results:
The Results section provides a clear and concise summary of the key findings, effectively presenting the data in a structured and informative manner. However, there is an area where the section could be improved: The authors could discuss the potential clinical implications of the findings, such as the importance of screening for autistic traits in young adults presenting with mental health symptoms.
Discussion:
The Discussion section provides a comprehensive analysis of the study findings and their implications. It effectively contextualizes the results within the broader literature and addresses potential limitations.
Still, there are a few areas where the section could be improved:
-Clinical Implications:
The authors could further elaborate on the clinical implications of their findings, such as the importance of early identification and intervention for young adults with autistic traits.
-Future Directions:
The discussion could suggest potential directions for future research to address the limitations of the current study and further explore the topic.
-Theoretical Implications:
The authors could discuss the implications of their findings for theoretical models of ASD and mental health.
Author Response
We thank the reviewer for their positive and helpful comments, which we hope have helped us improve our manuscript. For clarity, we addressed each comment one by one.
Reviewer 2
Investigating the prevalence of autistic traits in young adults seeking mental health services is essential for improving diagnosis, treatment, and overall well-being for individuals with ASD. By addressing the unique needs of this population, we can promote a more inclusive and supportive mental health care system. My suggestions for improving the manuscript are listed in the following:
Introduction: It effectively introduces the topic of autistic traits in young adults seeking mental health services, highlighting the key challenges and the importance of studying this population. The introduction is well-structured, concise, and engaging, making it a compelling starting point for the paper. However, I think the introduction could be revised based on the following considerations:
Comment: Review of the Literature: The literature cited could be up-to-date, especially considering the rapid advancements in research on ASD and mental health. Summarize the most relevant findings from previous studies to provide a solid foundation for the current research.
Response: Recent studies on ASD and mental health have explored the increasing overlap between ASD and various psychiatric comorbidities, contributing to a deeper understanding of its complexity.
To account for more recent findings, we added the following paragraphs with respect to ASD comorbidities: “Anxiety is another common comorbidity in individuals with ASD, with specific phobias and generalized anxiety disorder (GAD) being the most frequent. The prevalence of anxiety disorders overall in individuals with ASD is approximately 20-30. Moreover, recent research has drawn attention to the higher risk of suicide among individuals with ASD, especially whenanxiety, depression, or other psychiatric conditions are comorbid ultimately recommending close monitoring and mental health interventions to reduce the sucidal risk itself [21 -22].”
With respect to diagnostic methodology, the following paragraph was added, highlighting an important cross-cultural perspective: “Moreover, novel approaches in a cross cultural and personalized prospective have been explored. There is growing interest in validating diagnostic tools across different populations, as cultural factors may influence how autism is expressed and diagnosed. These approaches could enable more tailored treatments for ASD in the future”.
Relevant references were added.
Comment: Theoretical Framework: Discuss the theoretical frameworks (e.g., cognitive, social-emotional) that inform the research question. Clearly articulate how these theories relate to the study's objectives and contribute to our understanding of autistic traits in young adults.
Response: To better frame our study in the theory that supported our research question and hypothesis, first, we added the following paragraph at the beginning of the introduction: “The modern Dimensional Psychopathology Perspective considers ASD as existing on a continuum, rather than as a categorical disorder. […] Moreover, this perspective helps explaining why autistic traits might appear subthreshold but still contribute to significant impairments in functioning.”
Second, we wrote a single paragraph at the end of the discussion to address, in a synthetic way, the research gap regarding ASD assessment in young adults as follows: “To the best of our knowledge, no study has specifically examined the prevalence of ASD traits in a different clinical psychiatric sample of young adults, particularly those presenting with non-specific symptoms such as anxiety or depression. While autism research has traditionally centered on childhood diagnosis, far less is understood about young adults who may have undiagnosed ASD traits that only become evident when they face the heightened social and emotional demands of adulthood. Identifying autistic traits in this population could reveal cognitive processing difficulties that tend to emerge in social settings like work or higher education. These individuals may struggle to comprehend and interpret others' thoughts, emotions, and intentions, which is considered a core challenge in ASD and significantly impacts social communication and interaction. Understanding these dynamics helps explain the relationship between autistic traits and mental health challenges in young adults, supporting the need for early screening in clinical settings to better address the mental health needs of this population. Addressing this gap is crucial for both clinical practice and research, as it can prevent delayed or misdiagnosis and improve mental health outcomes and overall quality of life. Current screening practices often overlook ASD traits, particularly in patients without typical symptoms. Utilizing validated tools such as the AQ and the RAADS-R can enable earlier detection and lead to more comprehensive mental health assessments. Furthermore, given the comorbidities discussed earlier, identifying ASD traits as early as possible is especially important in young adults, as behavioral interventions applied early can reduce problematic behaviors in 80-90% of cases. Overall, closing this gap will enhance our understanding of how autistic traits manifest in adulthood and improve mental health care strategies for this group.”
Comment: Research Gap: Identify the specific knowledge gap that the study aims to address. Explain why filling this gap is crucial for advancing research and improving clinical practice.
Response: please see the answer above.
Comment: Research Questions or Hypotheses: State the research questions or hypotheses that the study will investigate. Ensure that the research questions or hypotheses are directly related to the topics discussed in the introduction.
Response: We moved and expanded our considerations about young adults in the introduction, and stated the aims of the study as follows: “The study’s main objective is to explore the prevalence of autistic traits in a clinical outpatient population of young adults aged 18 to 24 years old, using the two most common self-report screening questionnaires, namely the AQ and the RAADS-R. These questions aim to fill the knowledge gap concerning ASD in young adults with psychiatric conditions and contribute to more informed diagnostic and treatment practices. To better characterize our sample, we also investigated and discussed the role of sociodemographic factors such as sex assigned at birth and gender identity”
Comment: Significance: Emphasize the broader implications of the study's findings for individuals with ASD, mental health professionals, and policymakers.
Response: In the interest of synthesis, we addressed the significance and the potential benefits of our study in a single paragraph, as follows: “The study’s findings might lead to broader implications for both people with autism and mental health professionals. For individuals with ASD, early identification of autistic traits in young adults, particularly those with psychiatric conditions, can lead to improved mental health outcomes; as previously mentioned, young adults may experience significant distress from undiagnosed ASD traits, often leading to social, educational, and employment challenges. Mental health professionals could adopt in their initial screening validated tools as the AQ and RAADS-R to flag potential autistic traits that may otherwise go unnoticed. This could lead to more comprehensive treatment plans that address both ASD traits and co-occurring mental health issues, such as anxiety or depression. Additionally, understanding the relationship between sociodemographic factors and ASD traits allows for more sensitive, individualized care. Based on consisted literature, policy initiatives could emphasize the importance of early screening and intervention for ASD traits in adulthood. This includes promoting access to ASD diagnostic services in general mental health settings, creating employment support programs for individuals with ASD, and improving the accessibility of mental health services for marginalized groups. Policymakers can also advocate for expanded training programs for mental health professionals to better recognize and address autistic traits, thus reducing the rate of misdiagnosis and improving long-term outcomes for individuals with ASD [41]. These implications highlight the need for an integrated approach in both clinical practice and public health policy, focusing on early intervention and individualized support for those with ASD traits in adulthood.”
Relevant references were added.
Comment: Finally, I suggest highlighting potential benefits by discussing how the research may contribute to improved diagnosis, treatment, and outcomes for young adults with ASD.
Response: please see the answer above.
Methodology: the Materials and Methods section clearly and comprehensively describes the research procedures. It includes essential information on participants, procedures, and statistical analyses. However, there are a few areas where the section could be improved:
Comment: Ethical Considerations: While the authors mention obtaining informed consent, it would be beneficial to provide more details about the consent process, such as the information provided to participants and the measures taken to ensure their understanding. The section should explicitly address how participant confidentiality was maintained throughout the study.
Response: First, as stated in the exclusion criteria, if a participant was deemed by the clinician unable to understand the instruction of the task, they would be excluded from the study. To provide further information, we added the following paragraph to our method section: “Participants were explained in a clear language that the study consisted in compiling two self-report questionnaires, that would have been scored for both clinical and research purpose; it was made clear that, should the questionnaires indicated a presence of autistic traits, the participants would have been further evaluated with clinician-administered tool and further clinical interview; at the same time, it was explain that data would have been anonymized and shared amongst professionals bound by confidentiality within our hospital and always maintaining anonymity. It was stated that the study was approved by the local ethics committee and that participants, even after giving informed consent, were free to withdraw from the study at any time without giving further explanation”.
Comment: Clinical Assessment: This section could provide more information about the specific diagnostic criteria clinicians use to categorize participants' symptomatology. If multiple clinicians were involved in the assessments, the authors should discuss the measures taken to ensure inter-rater reliability.
Response: The following paragraph was added: “A thorough diagnostic assessment of general psychiatric conditions was conducted following DSM-5-TR criteria by a psychiatrist and a psychotherapist, by means of clinical interviews conducted after the first contact with the mental health service. Moreover, all cases, including the ones presented in this study, are usually discussed in Equipe Meetings, together with nurses, clinical psychologists, and social assistants”.
Comment: Statistical Power: While the sample size is relatively large, a power analysis could be conducted to determine the study's ability to detect significant effects.
Response: “Cohen’s D” values, reported in Table 2, represent the effect size for each t-test conducted to compare potential differences between males and females (sex assigned at birth) in the incidence of autistic traits. We gladly accept any other suggestion of potential analysis to be conducted to better demonstrate the statistical power of our results.
Comment: Missing Data: The authors should clarify how missing data were handled, such as imputation methods or exclusion criteria.
Response: We had no missing data with respect to the AQ and the RAADS-R. Participants who did not want to disclose their gender identity for personal reasons are grouped in the group “Undeclared”: it was treated not as a missing data, but as a specific and deliberate choice of the participant. We added this information in the manuscript.
Results: The Results section provides a clear and concise summary of the key findings, effectively presenting the data in a structured and informative manner. However, there is an area where the section could be improved.
Comment: The authors could discuss the potential clinical implications of the findings, such as the importance of screening for autistic traits in young adults presenting with mental health symptoms.
Response: The following sentence was added at the end of the results: “Overall, results show that screening tools such as the AQ and RAADS-R in this context provide valuable insights for clinicians, highlighting the importance of incorporating autism spectrum assessments in routine evaluations of young adults showing psychiatric symptoms.”
Discussion: The Discussion section provides a comprehensive analysis of the study findings and their implications. It effectively contextualizes the results within the broader literature and addresses potential limitations. Still, there are a few areas where the section could be improved:
Comment: Clinical Implications: The authors could further elaborate on the clinical implications of their findings, such as the importance of early identification and intervention for young adults with autistic traits.
Response: we added the paragraph: “Clinical implications, theoretical, and policy implications”, as follows: “The implications of our findings center on several key areas related to early identification and intervention for young adults with autistic traits. First, early detection can lead to timely interventions, potentially reducing the negative impact of untreated ASD symptoms. We advocate for the routine use of screening tools like the AQ and the RAADS-R in mental health settings to identify individuals who may have been overlooked in previous evaluations. Early identification is essential for designing personalized interventions and providing better access to re-sources, such as vocational training and social skills programs. These resources can help young adults to navigate critical life stages, ultimately preventing symptom escala-tion and improving overall quality of life and well-being. […] Finally, from a policy standpoint, our results emphasize the need for mental health systems to adopt more comprehensive screening protocols for autism in young adults presenting psychiatric symptoms. Spectrum-based approaches can guide the development of more inclusive diagnostic tools and interventions that address both autism and mental health needs. Creating supportive environments that recognize a spectrum of neurodiverse identities, both in everyday social contexts and mental health services, is crucial to ease the transition from pediatric to adult outpatient care.”
Comment: Future Directions: The discussion could suggest potential directions for future research to address the limitations of the current study and further explore the topic.
Response: the following paragraph was added: “Future research could aim to refine diagnostic tools such as the AQ (which is validated on a sample of male individuals only) and the RAADS-R for specific populations (e.g., women, people with psychiatric comorbidities). Developing more sensitive and specific screening measures for populations that often go undiagnosed could help bridging current gaps in clinical practice. An essential point should be testing the reliability of these results using other tools administrated by trained and expert clinicians. Second, a follow-up with clinician-administer tools such as the ADOS-2 would be crucial to test the incidence of ASD-diagnosed people in this wide group of individuals with autistic traits. Finally, future research would benefit from longitudinal designs: by following individuals over time, researchers can gain insight into how autistic traits and psychiatric symptoms co-evolve and how effective early interventions are.”
Comment: Theoretical Implications: The authors could discuss the implications of their findings for theoretical models of ASD and mental health.
Response: Executive functioning and social cognition are key areas of research in understanding ASD; however, our results are too preliminary to lead to a significant change in validated theoretical model. We briefly addressed the suggested topic as follows, in the new paragraph discussing clinical, theoretical and policy implication: “Second, from a theoretical perspective, identifying psychiatric symptoms co-occurring with autistic traits can refine our understanding of how neurodevelopmental and psychiatric conditions interact. For instance, these overlapping symptoms may suggest that, in a transitional life stage like young adulthood, difficulties in executive function might exacerbate psychiatric conditions”.
Round 2
Reviewer 2 Report
Comments and Suggestions for Authors
This updated version has been improved. I appreciate the authors' effort in addressing the comments and suggestions I provided.